# Novel Anthraquinone-Based Benzenesulfonamide Derivatives and Their Analogues as Potent Human Carbonic Anhydrase Inhibitors with Antitumor Activity: Synthesis, Biological Evaluation, and In Silico Analysis

**DOI:** 10.3390/ijms25063348

**Published:** 2024-03-15

**Authors:** Shanshan Wu, Xiaoping Zhou, Fei Li, Wei Sun, Qingchuan Zheng, Di Liang

**Affiliations:** School of Pharmaceutical Sciences, Jilin University, Changchun 130021, China; wuss21@mails.jlu.edu.cn (S.W.); zhouxp@jlu.edu.cn (X.Z.); lif23@mails.jlu.edu.cn (F.L.); wsun@jlu.edu.cn (W.S.); zhengqc@jlu.edu.cn (Q.Z.)

**Keywords:** anthraquinone, carbonic anhydrase inhibitors, molecular docking, SLC-0111, acetazolamide

## Abstract

In this study, we designed two series of novel anthraquinone-based benzenesulfonamide derivatives and their analogues as potential carbonic anhydrase inhibitors (CAIs) and evaluated their inhibitory activities against off-target human carbonic anhydrase II (hCA II) isoform and tumor-associated human carbonic anhydrase IX (hCA IX) isoform. Most of these compounds exhibited good inhibitory activities against hCA II and IX. The compounds that exhibited the best hCA inhibition were further studied against the MDA-MB-231, MCF-7, and HepG2 cell lines under hypoxic and normoxic conditions. Additionally, the compounds exhibiting the best antitumor activity were subjected to apoptosis and mitochondrial membrane potential assays, which revealed a significant increase in the percentage of apoptotic cells and a notable decrease in cell viability. Molecular docking studies were performed to demonstrate the presence of numerous hydrogen bonds and hydrophobic interactions between the compounds and the active site of hCA. Absorption, distribution, metabolism, excretion (ADME) predictions showed that all of the compounds had good pharmacokinetic and physicochemical properties.

## 1. Introduction

Anthraquinones (anthracene-9,10-dione; Figure 1) are important organic compounds with a wide range of applications. More than 75 natural anthraquinone compounds have been discovered from various sources, including bryophytes, marine resources, fungi, and medicinal plants from different families [1,2]. In traditional medicinal herbs, anthraquinone compounds have been demonstrated to be important active ingredients and have been used for more than 4000 years [3]. Anthraquinone compounds exhibit a wide range of pharmacological activities, including antitumor, antibacterial, anti-inflammatory, antioxidant, antidiabetic, antiviral, hepatobiliary, visual, antiaging, anti-mutation, anti-ultraviolet, sleep-promoting, and hypotensive therapeutic applications. Many natural anthraquinone compounds, such as emodin, aloe emodin, and rhein, are often used as starting materials for developing antitumor drugs [4,5]. In addition, some formulations containing anthraquinone skeletons, such as epirubicin [6] and mitoxantrone [7], have been approved for clinical use (Figure 1).

Carbonic anhydrases (CAs; EC4.2.1.1) are metalloenzymes that exist in most living organisms. CAs catalyze the mutual conversion between CO_2_ and bicarbonate, and they participate in the regulation of pH, lipogenesis, ureagenesis, gluconeogenesis, and tumorigenicity [8]. There are eight genetically unrelated known families of CA_S_ (*α*-, *β*-, *γ*-, *δ*-, *ζ*-, *η*-, *θ*-, and *i*-CAs), and the last three have only recently been discovered [9]. It should be noted that human CAs (hCAs) belong to the *α*-class and have 16 different isoforms [10]. Among the isoforms, 12 hCA isoforms (CA I−IV, VA−VB, VI−VII, IX, and XII−XIV) show varying degrees of enzymatic activity, while 3 hCA isoforms (VIII, X, and XI) show no catalytic activity [11]. Research has shown that many human diseases are closely associated with abnormal levels of these enzymes. Modulation of CA activity via CA inhibitors (CAIs) has been a validated clinical strategy for decades. For example, inhibitors targeting hCA I are utilized for retinal and cerebral edema, while inhibitors targeting hCA II are used as diuretics and anti-glaucoma agents [12], and hCA II is widely associated with various types of cancers, with a recent study finding that it is expressed in the endothelium of neovessels in melanoma, as well as esophageal, renal, and lung cancers [13]. In addition, hCA IX is related to cancer development; hCA IX is not highly expressed in most normal tissues but is abundant in gastric and gallbladder epithelial cells [14], and it is highly expressed in various solid tumors [15,16]. The role of hCA IX as a regulatory enzyme in tumors suggests its potential as an anticancer drug target [17]. Although many hCA IX inhibitors have been reported, only one small-molecule drug, namely, SLC-0111 (Figure 1), is in clinical phase I/II trials.

SLC-0111 is a benzenesulfonamide-based hCA IX inhibitor that is currently in clinical trials for the treatment of advanced metastatic hypoxic tumors. Research has been focused on the structural modification of SLC-0111, as a lead molecule, to identify new effective inhibitors for tumor-related hCA IX [18]. There are several strategies for the structural modification of SLC-0111. Firstly, the sulfonamide moiety in SLC-0111 is an important group because the sulfonamide has a zinc-binding group (ZBG) that binds to Zn^2+^ at the active site. Other studies have used heterocyclic sulfonamides, carboxylic acids, or esters instead of sulfonamides to prepare selective and potent scaffolds for hCA IX inhibitors [19]. Secondly, studies have exploited several SLC-0111 analogues as effective inhibitors for the hCA IX isoform through the replacement of 4-fluorophenyl as a tail in SLC-0111 with different groups, such as thiazole [20] and triazine [21], with some analogues displaying better hCA IX-inhibitory effects than SLC-0111. Thirdly, the ureido linker in SLC-0111 provides flexibility in the tail of the molecule, allowing the inhibitor to adopt a variety of orientations. Replacing the ureido linker with a thioureido or acetamido linker has also been a common method of structural modification in recent years. For example, compound **I** and compound **II** have been reported as highly potent and selective inhibitors of hCA IX [22,23]. In addition, the structure of acetazolamide (AAZ; Figure 1), which contains an acetamido group as a linker, has been used as a classic CA inhibitor.

In recent years, some studies have also introduced natural active molecules into SLC-0111 analogues as tails (Figure 1), such as natural piperine moiety **III** and ligustrazine moiety **IV** [24,25], resulting in excellent selectivity and inhibitory activity. However, anthraquinone-based SLC-0111 analogues have not been reported. In this article, we report for the first time the design and synthesis of carbonic anhydrase inhibitors containing anthraquinone skeletons. We selected two isoforms with different pharmacological activities (off-target hCA II and tumor-related hCA IX) for the evaluation of their enzymatic activities. By studying the inhibitory activity and antitumor activity, we aimed to preliminarily understand the pharmacological advantages of inhibitors with this type of skeleton.

In the present study, through the modification of SLC-0111, two series of anthraquinone derivatives and analogues were designed and developed. Firstly, the 4-fluorophenyl tail of SLC-0111 was replaced with the anthraquinone moiety (pink in Figure 2), and the ureido linker in SLC-0111 was replaced with a carbonyl thioureido moiety (green in Figure 2). Subsequently, further modifications or replacements were made to the benzenesulfonamide moiety (blue in Figure 2) in the novel anthraquinone compounds after replacing the tail and linker of SLC-0111, in order to obtain the first series of novel CA inhibitors (**5a**–**n**) (Figure 2). By replacing the carbonyl thioureido linker in the first series of compounds with an acetamido moiety (green in Figure 2), the second series of novel CA inhibitors (**6a**–**e**, **h**–**k**) were obtained (Figure 2).

## 2. Results and Discussion

### 2.1. Chemistry

The preparation of anthraquinone-carbonyl thioureido sulfonamide derivatives and analogues (compounds **5a**–**n**) as well as anthraquinone-acetamido sulfonamide derivatives and analogues (compounds **6a**–**e** and **6h**–**k**) reported in this study is illustrated in Figure 1 and Figure 2, respectively.

For Figure 1, anthraquinone-2-carboxylic acid (**1**) was refluxed under the addition of thionyl chloride to yield anthraquinone-2-formyl chloride (**2**). After removing the thionyl chloride under vacuum, **2** was refluxed in acetone with the presence of potassium thiocyanate to produce the key intermediate anthraquinone-2-formyl isothiocyanate (**3**). Intermediate **3** was reacted with various anilines attached to different metal-binding groups in an acetone solution to yield compounds **5a**–**n** (yields of 63–84%).

For Figure 2, the intermediate anthraquinone-2-formyl chloride (**2**) synthesized above was subjected to amide condensation reactions with various anilines connected with different metal-binding groups to synthesize compounds **6a**–**e** and **6h**–**k** (yields of 59–80%).

The ^1^H NMR, ^13^C NMR, and HRMS techniques were used for the characterization of the newly synthesized compounds.

The structures of 23 newly synthesized compounds were in agreement with their spectral analysis data (Appendix A). It should be emphasized that in hydrogen spectroscopy analysis, we found that the presence of labile NH or COOH protons in the synthesized compound resulted in the absence of H protons in the ^1^H NMR spectrum. The summary is as follows: Compounds **5a**–**b**, **g**–**h,** and **m** were missing one H proton from -CONHCS-. Compounds **5c**–**f** were missing two H protons from -NHCSNH. Compound **5l** was missing two H protons from -NHCSNH- and one H proton from -SO_2_NH-. Compounds **6a**–**b** were missing one H proton from -COOH. Compound **6k** was missing one H proton from SO_2_NH-.

### 2.2. Biological Evaluation

#### 2.2.1. Carbonic Anhydrase Inhibition

All of the newly synthesized compounds, including the anthraquinone-carbonyl thioureido sulfonamide derivatives and analogues (compounds **5a**–**n**), as well as the anthraquinone-acetamido sulfonamide derivatives and analogues (compounds **6a**–**e** and **6h**–**k**), were evaluated for inhibitory activity against the off-target hCA II and tumor-associated *h*CA IX isoforms. The CA-inhibitory activities were compared to **AAZ** as a positive control. The inhibitory activity was evaluated by monitoring the hydrolysis of 4-nitrophenylacetate (4-NPA.). The inhibition data are summarized in Table 1.

As shown in Table 1, the structure–activity relationships (SARs) of all compounds were preliminarily analyzed.

The inhibitory concentrations of all compounds were in the nanomolar range. Some of the synthesized sulfonamide compounds showed significant activity against both off-target hCA II and tumor-associated hCA IX, comparable to the activity of the **AAZ** positive control. Replacing the sulfonamide group with carboxylic acid (compounds **5a**–**b** and **6a**–**b**), ketone (compounds **5e** and **6e**), and ester (compounds **5f** and **5g**) groups significantly reduced the inhibitory activity, indicating that the sulfonamide group was the better ZBG.

The SAR of hCA II showed that compounds with a primary sulfonamide-binding group had better activity (IC_50_ range: 21.19–91.16 nM). Non-sulfonamide compounds, such as compounds **5a**–**b**, **5e**–**g**, **6a**–**b**, and **6e**, displayed moderate inhibitory activity, with IC_50_ values in the high nanomolar range (113.08–295.94 nM). Additionally, the inhibitory activities against hCA II were reduced when the carbonyl thioureido linker in compounds **5c**, **5d**, and **5h** was substituted with an acetamido linker to obtain compounds **6c**, **6d**, and **6h**, respectively, indicating that carbonyl thioureido may be a better linker for hCA II inhibitors. Interestingly, the highest hCA II-inhibitory activity in the two series of inhibitors was achieved by the *para*-benzenesulfonamide-bearing counterparts in compounds **5c** and **5h** (IC_50_ = 41.58 and 21.19 nM, respectively; carbonyl thioureido linker series), as well as in compounds **6c** and **6h** (IC_50_ = 68.34 and 61.76 nM, respectively; acetamido linker series).

Regarding the inhibitory impact of the prepared anthraquinone-based sulfonamides on the tumor-related hCA IX, all of the tested sulfonamide derivatives inhibited this isoform, with inhibition constants in the sub-micromolar range (30.06–274.81 nM). Similar to the SAR of hCA II, the highest hCA IX-inhibitory activity was also achieved by the *para*-sulfonamide substituent in compounds **5c**, **5h**, **6c**, and **6h** (IC_50_ = 40.58, 46.19, 34.88, and 30.06 nM, respectively). Compounds **5c**, **5h**, **6c**, and **6h** also exerted better activity than the **AAZ** positive control (IC_50_ = 49.31 nM). Further SAR analysis revealed that meta-regioisomers displayed much weaker potencies against hCA IX than their para-analogues in both carbonyl thioureido sulfonamides (**5d**; IC_50_ = 73.56 nM vs. **5c**; IC_50_ = 40.58 nM) and acetamido sulfonamides (**6d**; IC_50_ = 55.87 nM vs. **6c**; IC_50_ = 34.88 nM). The inhibitory activities against hCA IX increased when the carbonyl thioureido linker in compounds **5c**, **5d**, and **5h** was substituted with an acetamido linker to obtain compounds **6c**, **6d**, and **6h**, respectively.

Interestingly, elongation of the carbonyl thioureido linker in the para-regioisomer **5c** with an ethylene (-CH_2_CH_2_-) spacer (compound **5h**) led to increased potency toward the hCA II isoform, whereas the inhibitory activity against the hCA IX isoform was decreased with this elongation approach. In contrast, elongation of the linker in amide-based **6c** improved the inhibitory activity of all of the examined isoforms. In addition, substitution of the sulfonamide amino group reduced the CA-inhibitory activity.

#### 2.2.2. Anticancer Activity

##### In Vitro Anti-Proliferative Activity

The CA assay (Table 1) showed that compounds **5c**, **5h**, **6c**, and **6h** were efficient hCA IX inhibitors (IC_50_ = 40.58, 46.19, 34.88, and 30.06 nM, respectively). Therefore, the potential anticancer properties of these small molecules were further investigated. The anti-proliferative activities of compounds **5c**, **5h**, **6c**, and **6h** were examined by an MTT assay after treatment of MDA-MB-231, MCF-7 and HepG2 cancer cells for 24 h. The median growth-inhibitory concentration (IC_50_) in the treated cells was compared to that of untreated controls. Cells were cultured under normoxic and hypoxic conditions to evaluate the effects of the compounds on cancer cell viability. Staurosporine was utilized as the positive control at the same concentrations. The obtained results are shown in Table 2 and Figure 3.

As displayed in Table 2 and Figure 3, only compound **5c** showed moderate anti-proliferative activity against the MDA-MB-231, MCF-7, and HepG2 cancer cell lines, with IC_50_ values of 48.10 ± 3.23 μM, 34.33 ± 2.85 μM, and 61.36 ± 5.75 μM under normoxic conditions and 46.23 ± 3.93 μM, 28.12 ± 1.72 μM, and 69.43 ± 5.16 μM under hypoxic conditions, respectively, compared to the reference drug staurosporine, which had IC_50_ values of 10.30 ± 0.51μM, 7.81 ± 0.59 μM, and 18.71 ± 1.45 μM under normoxic conditions and 10.55 ± 0.84 μM, 7.65 ± 0.73 μM, and 17.78 ± 0.97 μM under hypoxic conditions, respectively. Compared to compound **5c**, compounds **5h**, **6c**, and **6h** showed weaker inhibitory activity against tumor cells. Although compound **6h** had the highest hCA IX-inhibitory activity, it did not have the highest antitumor activity in the MDA-MB-231 and MCF-7 cancer cell lines, indicating that hCA IX is a tumor-associated target and that hCA IX inhibitors are not sufficiently potent to be used alone in cancer therapy [26]. Under hypoxic conditions, the inhibitory activity of the selected compounds against the MCF-7 cell line was slightly increased. As previously reported, the expression of hCA IX is enhanced under hypoxic conditions in this cell type [27]. Therefore, compounds with good inhibitory activity against hCA IX anhydrase may exhibit better inhibitory activity under hypoxic conditions. 

##### Annexin V-FITC/Propidium Iodide Analysis of Apoptosis

Annexin V-FITC/propidium iodide analysis of apoptosis assays were conducted to evaluate the effect of 5c on apoptosis in the MDA-MB-231, MCF-7, and HepG2 cancer cell lines. As shown in Table 3 and Figure 4, a significant increase in the percentage of apoptotic cells (containing early and late apoptosis percentages) and a notable decrease in cell viability were observed after incubation with compound **5c** for 24 h.

##### Mitochondrial Membrane Potential Analysis

The present study investigated the involvement of mitochondria in compound-5-induced apoptosis by JC-1 staining. Red fluorescence indicates JC-1 aggregates, and green fluorescence indicates membrane potential leading to an increase in JC-1 monomers. After 48 h of treatment with compound **5c** at a concentration of 20 μM, a reduction in mitochondrial membrane potential was observed in the MDA-MB-231, MCF-7, and HepG2 cell lines. As shown in Figure 5, the green fluorescence was significantly enhanced after treatment with compound **5c** in the MDA-MB-231, MCF-7, and HepG2 cell lines, suggesting that compound **5c** induced JC-1 monomer accumulation, mitochondrial membrane potential alteration, and cell damage. These results were consistent with the Hoechst 33342 nuclear staining. Together, these results demonstrate that compound **5c** induces cell apoptosis. 

### 2.3. Molecular Modeling Studies

To validate the docking protocol, the original ligands from the crystal structures were re-docked, and the root-mean-square deviation (RMSD) was calculated through PyMOL. The results indicated that the RMSD values between the docked conformation of the original ligands from the crystal structures and the corresponding crystal structures were 0.523 Å and 0.399 Å for hCA II (PDB ID: 5GMN) and hCA IX (PDB ID: 5FL4), respectively, both of which were less than 1 Å (Appendix A). Among the synthesized anthraquinone-based CAIs, compounds **5c**, **5h**, **6c**, and **5k** were subjected to docking studies to reveal the relationships between structural features and inhibitory profiles towards the off-target hCA II (Figure 6) and the tumor-related isoform hCA IX (Figure 7). 

Within the binding site of the tumor-associated hCA II, the sulfonamide of the compounds fits deeply inside the shallow CAIX active site, anchoring the zinc ion via a metal bond with Zn^2+^, which is typical of sulfonamide CAIs. The oxygen of the sulfonamide in compounds **5c**, **5h**, and **6c** showed a hydrogen bond interaction with the peptide nitrogen of the T198 residue. The oxygen on the anthraquinone of compound **5h** formed a hydrogen bond with N67. The anthraquinone rings of compounds **5c**, **5h**, and **6c** engaged in pi-stacking interactions with F130, which were further stabilized by hydrophobic interactions with residues L60 and E69 in the hCA II-binding pocket.

Regarding hCA IX, a similar set of interactions of compounds **5c**, **5h**, and **6c** in the active site of hCA IX exhibited the typical metal interaction of the sulfonamide group with the Zn metal (Figure 7). The NH- of sulfonamide (compounds **5c**, **5h**, and **6c**) displayed hydrogen bonding with the E106 amino acid. The oxygen of sulfonamide in compounds **5c**, **5h**, and **6c** showed a hydrogen bond interaction with the peptide nitrogen of the T200 residue. The oxygen and nitrogen of the carbonyl thioureido linker were stabilized by a hydrogen bond interaction with W9 and P202, respectively, in compounds **5c** and **5h**. Moreover, hydrogen bonding interactions between the oxygen of the amidic group in compound **6c** and the Q92 residue were also found to involve the stabilization of the ligand–CA IX complex. In addition, compounds **5c**, **5h**, and **6c** formed hydrophobic interactions with different amino acid residues.

Although most of the compounds showed good inhibitory activity, some of them showed weak activity. To explain this phenomenon, compound **5k** was docked with hCA II and hCA IX. As shown in Figure 6 and Figure 7, compound **5k** underwent considerable torsion, which led to an increase in the distance between the nitrogen atom of the sulfonamide group and the zinc ion. The increased distance was attributed to the poor activities of this compound. The docking scores (Kcal/mol), detailed binding interactions, and IC_50_ values of the most active compounds with inhibitory activity against the hCA II and hCA IX isozymes are depicted in Appendix A.

### 2.4. ADME Prediction

Many potential enzyme inhibitors have poor pharmacokinetic properties, making it difficult for them to reach clinical trials. To understand their basic properties, it is necessary to study the pharmacokinetics of compounds. In the present study, SwissADME online software (http://www.swissadme.ch, accessed on 20 November 2023) was used to predict the ADME for all compounds. Figure 8 shows the BOILED-Egg plot of the WLOGP vs. topological polar surface area (TPSA) of the tested compounds. As shown in Figure 8, all of the compounds showed no permeability of the blood–brain barrier (BBB), indicating their safety for the central nervous system (CNS). None of the compounds tested were P-glycoprotein substrates (PGP-), and none of the compounds were affected by the efflux process of transporters, which is a resistance mechanism in many cancer cell lines, indicating their potential safety for cancer treatment. Compounds **5e**–**g**, **6a**–**e**, and **6h** had high gastrointestinal tract (GIT) absorption (oral bioavailability), whereas the other compounds had lower GIT absorption. Compounds **5e**–**g**, **6a**–**e**, and **6h** also showed higher human intestinal absorption (HIA) areas. Except when the values of POLAR for compounds **5a**–**d**, **5h**–**n**, and **6j**–**k** were not within the standard range, all tested compounds satisfied the observations of Lipinski (Pfizer) and Ghose (Amgen), and all parameter values were within the standard ranges. Computational analysis of ADME showed that all compounds followed Lipinski’s ‘Rule of five’, and most of the compounds exhibited potential bioavailability and pharmacokinetic profiles.

## 3. Materials and Methods

### 3.1. Chemistry

#### 3.1.1. General

The commercial chemicals utilized herein were of reagent grade and were used without further purification. The solvent was treated with an activated molecular sieve for anhydrous treatment. Analytical thin-layer chromatography (TLC) was performed using silica gel HF_254_ (Qingdao Hai Yang Chemical, Qingdao, China). An SGW X-4B micro-melting point instrument was employed to measure the melting points of the synthesized compounds, which were uncorrected. The ^1^H-NMR and ^13^C-NMR spectra were recorded on a Varian (300 MHz) spectrometer or a Bruker (600 MHz) spectrometer, using DMSO-*d*_6_ as a solvent. The splitting abbreviations were as follows: s, singlet; d, doublet; dd, doublet of doublets; t, triplet; m, multiplet. The Orbitrap Exploris 480 mass spectrometer (Thermo Scientific, Bremen, Germany) was employed to conduct high-resolution mass spectrometry (HRMS). The SIL-20A high-performance liquid chromatograph (HPLC, Shimadzu, Japan) was used to assess the optical purity of the synthesized compounds **5c**, **5h**, **6c**, and **6h**: Shim-pack GIST C18 column, water/methanol = 40:60, 1.0 mL/min and λ = 254 nm. All of the analyzed compounds were >95% HPLC pure (Appendix A).

#### 3.1.2. Synthesis of Anthraquinone-2-Formyl Chloride (**2**)

First, 0.252 g (1 mmol) of anthraquinone-2-carboxylic acid (**1**) and thionyl chloride (5 mL) were added to a 3-necked flask and refluxed at 85 °C. The reaction was monitored by TLC and completed in 2 h. The mixture was cooled to room temperature, and the remaining thionyl chloride was evaporated to generate anthraquinone-2-formyl chloride (**2**) [28].

#### 3.1.3. Synthesis of Anthraquinone-2-Formyl Isothiocyanate (**3**)

A solution of 0.271 g (1 mmol) anthraquinone-2-formyl chloride (**2**) in acetone (27 mL) was added dropwise to a suspension of potassium thiocyanate (1 mmol) in acetone (41 mL), refluxing at 57 °C. The reaction was monitored by TLC and completed in 1.5 h, producing an isothiocyanate salt and forming the intermediate anthraquinone-2-formyl isothiocyanate (**3**) in the reaction medium.

#### 3.1.4. General Procedure for Target Compounds **5a**–**n**

Compounds **4a**–**n** (1 mmol) were added to the acetone solution of anthraquinone-2-formyl isothiocyanate (**3**), and the mixture was refluxed and stirred for 4–8 h. After the reaction was completed (monitored by TLC), the resulting precipitate was collected by filtration and recrystallized with dimethylformamide to obtain the pure products **5a**–**n**.

##### (3-(9,10-Dioxo-9,10-Dihydroanthracene-2-Carbonyl) Thioureido) Benzoic Acid (**5a**)

Orange solid (yield 81%), m.p. 221.0–221.4 °C; ^1^H NMR (300 MHz, DMSO-*d*_6_) *δ* 12.57 (s, 2H), 8.70 (d, *J* = 1.8 Hz, 1H), 8.41 (dd, *J* = 8.1, 1.9 Hz, 1H), 8.31 (d, *J* = 8.1 Hz, 1H), 8.28–8.20 (m, 2H), 8.02–7.97 (m, 2H), 7.97–7.92 (m, 2H), 7.92–7.87 (m, 2H) ppm; ^13^C NMR (75 MHz, DMSO-*d*_6_) *δ* 182.0, 181.8, 178.8, 166.8, 166.7, 141.9, 137.2, 135.6, 134.8, 134.1, 133.1, 133.0, 132.7, 130.0, 128.2, 127.4, 127.0, 126.9, 123.9, 123.5 ppm; HRMS (+) *m*/*z* [M + H]^+^ 431.0702 (calcd for C_23_H_15_N_2_O_5_S, 431.0693).

##### 3-(3-(9,10-Dioxo-9,10-Dihydroanthracene-2-Carbonyl) Thioureido) Benzoic Acid (**5b**)

Faint yellow solid (yield 72%), m.p. 221.0–221.3 °C; ^1^H NMR (300 MHz, DMSO-*d*_6_) *δ* 12.45 (s, 2H), 8.71 (d, *J* = 1.9 Hz, 1H), 8.42 (dd, *J* = 8.1, 1.9 Hz, 1H), 8.35–8.29 (m, 2H), 8.29–8.24 (m, 1H), 8.24–8.20 (m, 1H), 7.97 (dd, *J* = 5.8, 3.3 Hz, 2H), 7.93–7.81 (m, 2H), 7.55 (t, *J* = 7.9 Hz, 1H) ppm; ^13^C NMR (75 MHz, DMSO-*d*_6_) *δ* 182.0, 181.9, 179.3, 167.5, 166.8, 138.3, 137.3, 135.6, 134.8, 134.1, 133.1, 133.0, 132.8, 132.7, 131.3, 129.0, 128.8, 128.1, 127.4, 127.1, 126.9, 125.4, 125.3 ppm; HRMS (+) *m*/*z* [M + H]^+^ 431.0702 (calcd for C_23_H_15_N_2_O_5_S, 431.0689).

##### 9,10-Dioxo-*N*-((4-Sulfamoylphenyl)Carbamothioyl)-9,10-Dihydroanthracene-2-Carboxamide (**5c**)

Yellow solid (yield 76%), m.p. 228.0–228.6 °C; ^1^H NMR (300 MHz, DMSO-*d*_6_) *δ* 8.70 (d, *J* = 1.8 Hz, 1H), 8.41 (dd, *J* = 8.1, 1.9 Hz, 1H), 8.32 (d, *J* = 8.1Hz, 1H), 8.28–8.21 (m, 2H), 7.98 (q, 2H), 7.96–7.92 (m, 2H), 7.90 (d, *J* = 7.5 Hz, 2H), 7.86 (d, *J* = 2.1 Hz, 1H), 7.43 (s, 1H) ppm; ^13^C NMR (75 MHz, DMSO-*d*_6_) *δ* 182.0, 181.8, 179.2, 166.7, 141.4, 140.8, 137.1, 135.6, 134.8, 134.1, 133.0, 133.0, 132.7, 127.5, 127.4, 126.9, 126.3, 124.4, 120.2 ppm; HRMS (+) *m*/*z* [M + H]^+^ 466.0531 (calcd for C_22_H_16_N_3_O_5_S_2_, 466.0522).

##### 9,10-Dioxo-*N*-((3-Sulfamoylphenyl)Carbamothioyl)-9,10-Dihydroanthracene-2-Carboxamide (**5d**)

Faint yellow solid (yield 84%), m.p. 146.5–146.8 °C; ^1^H NMR (300 MHz, DMSO-*d*_6_) *δ* 8.71 (d, *J* = 1.9 Hz, 1H), 8.42 (dd, *J* = 8.2, 1.9 Hz, 1H), 8.33 (d, *J* = 8.1 Hz, 1H), 8.26 (m, 2H), 8.22 (d, *J* = 6.4 Hz, 1H), 7.97 (dd, *J* = 5.8, 3.3 Hz, 2H), 7.92 (s, 1H), 7.74 (d, *J* = 7.8 Hz, 1H), 7.64 (t, *J* = 7.9 Hz, 1H), 7.49 (s, 1H) ppm; ^13^C NMR (75 MHz, DMSO-*d*_6_) *δ* 182.0, 181.8, 179.3, 166.7, 144.5, 138.4, 137.2, 135.5, 134.8, 134.1, 134.0, 133.1, 133.0, 132.7, 129.4, 127.7, 127.4, 127.2, 123.4, 121.6, 116.1 ppm; HRMS (+) *m*/*z* [M + H]^+^ 466.0531 (calcd for C_22_H_16_N_3_O_5_S_2_, 466.0519).

##### ((4-Acetylphenyl)Carbamothioyl)-9,10-Dioxo-9,10-Dihydroanthracene-2-Carboxamide (**5e**)

Faint yellow (yield 70%). m.p. 215.0–215.9 °C; ^1^H NMR (300 MHz, DMSO-*d*_6_) *δ* 8.70 (d, *J* = 1.9 Hz, 1H), 8.41 (dd, *J* = 8.1, 1.9 Hz, 1H), 8.31 (d, *J* = 8.1 Hz, 1H), 8.24 (m, 2H), 8.07–7.99 (m, 2H), 7.97 (d, *J* = 3.3 Hz, 1H), 7.95 (q, *J* = 2.5, 2.1 Hz, 2H), 7.92 (t, *J* = 3.3 Hz, 1H), 2.59 (s, 3H) ppm; ^13^C NMR (75 MHz, DMSO-*d*_6_) *δ* 196.8, 182.0, 181.8, 178.8, 166.8, 142.1, 137.2, 137.2, 134.8, 134.1, 133.3, 133.1, 132.8, 132.7, 130.5, 128.9, 127.4, 127.1, 126.9, 123.4, 123.3, 26.7 ppm; HRMS (+) *m*/*z* [M + H]^+^ 429.0909 (calcd for C_24_H_17_N_2_O_4_S, 429.0900).

##### Methyl 4-(3-(9,10-Dioxo-9,10-Dihydroanthracene-2-Carbonyl)Thioureido)Benzoate (**5f**)

Faint yellow (yield 63%), m.p. 211.0–211.4 °C; ^1^H NMR (300 MHz, DMSO-*d*_6_) *δ* 8.69 (dd, *J* = 11.1, 1.8 Hz, 1H), 8.41 (m, 1H), 8.35–8.24 (m, 2H), 8.24–8.19 (m, 1H), 8.04–7.96 (m, 2H), 7.96–7.88 (m, 2H), 7.69–7.63 (m, 1H), 6.66–6.59 (m, 1H), 3.84 (d, *J* = 13.4 Hz, 3H) ppm; ^13^C NMR (75 MHz, DMSO-*d*_6_) *δ* 182.1, 181.9, 179.8, 165.9, 165.9, 135.6, 134.7, 134.4, 134.0, 133.2, 133.0, 132.7, 132.4, 130.6, 130.3, 127.4, 126.9, 126.6, 124.1 124.0, 116.8, 115.1, 51.4 ppm; HRMS (+) *m*/*z* [M + H]^+^ 445.0858 (calcd for C_24_H_17_N_2_O_5_S, 445.0844).

##### Methyl 3-(3-(9,10-Dioxo-9,10-Dihydroanthracene-2-Carbonyl)Thioureido)Benzoate (**5g**)

Faint yellow solid (yield 67%), m.p. 187.0–187.3 °C; ^1^H NMR (300 MHz, DMSO-*d*_6_) *δ* 8.70 (d, *J* = 1.8 Hz, 1H), 8.41 (dd, *J* = 8.1, 1.9 Hz, 1H), 8.35 (s, 1H), 8.29 (d, *J* = 7.9 Hz, 1H), 8.27–8.20 (m, 2H), 8.00–7.95 (m, 2H), 7.95 (s, 2H), 7.86 (m, 1H), 7.59 (t, *J* = 7.9 Hz, 1H), 3.88 (s, 3H) ppm; ^13^C NMR (75 MHz, DMSO-*d*_6_) *δ* 182.0, 181.8, 179.3, 165.7, 162.3, 138.5, 137.2, 135.5, 134.8, 134.4, 134.1, 133.1, 132.7, 130.1, 130.0, 129.2, 127.4, 127.3, 126.9, 126.7, 125.1, 125.1, 52.4 ppm; HRMS (+) *m*/*z* [M + H]^+^ 445.0858 (calcd for C_24_H_17_N_2_O_5_S, 445.0846).

##### 9,10-Mioxo-*N*-((4-Sulfamoylphenethyl)Carbamothioyl)-9,10-Dihydroanthracene-2-Carboxamide (**5h**)

Yellow solid (yield 68%), m.p. 323.0–323.5 °C; ^1^H NMR (300 MHz, DMSO-*d*_6_) *δ* 9.09 (t, *J* = 5.5 Hz, 1H), 8.62 (dd, *J* = 1.6, 0.8 Hz, 1H), 8.29 (t, *J* = 1.4 Hz, 2H), 8.26–8.19 (m, 2H), 7.98–7.92 (m, 2H), 7.79–7.73 (m, 2H), 7.50–7.43 (m, 2H), 7.30 (s, 2H),3.59 (q, *J* = 6.7 Hz, 2H), 2.98 (t, *J* = 7.1 Hz, 2H) ppm; ^13^C NMR (75 MHz, DMSO-*d*_6_) *δ* 182.1, 182.1, 182.0, 164.6, 143.7, 142.1, 139.2, 134.7, 134.6, 134.6, 133.1, 133.0, 133.0, 132.7, 129.2, 127.1, 126.8, 126.8, 125.7, 125.4, 40.6, 34.6 ppm; HRMS (+) *m*/*z* [M + H]^+^ 594.0844 (calcd for C_24_H_19_N_3_O_5_S_2_, 494.0833).

##### ((4-(*N*-Methylsulfamoyl)Phenyl)Carbamothioyl)-9,10-Anthraquinone-2-Carboxamide (**5i**)

Faint yellow solid (yield 72%), m.p. 207.0–207.6 °C; ^1^H NMR (300 MHz, DMSO-*d*_6_) *δ* 12.59 (s, 1H), 12.24(s, 1H), 8.73–8.69 (m, 1H), 8.41 (dd, *J* = 8.1, 1.9 Hz, 1H), 8.36–8.29 (m, 1H), 8.28–8.20 (m, 2H), 8.00 (d, *J* = 2.0 Hz, 1H), 7.99–7.97 (m, 2H), 7.96 (d, *J* = 3.4 Hz, 1H), 7.86–7.80 (m, 2H), 7.50 (q, *J* = 5.0 Hz, 1H), 2.45 (d, *J* = 5.1 Hz, 3H) ppm; ^13^C NMR(75 MHz, DMSO-*d*_6_) *δ* 182.0, 181.8, 179.1, 166.7, 141.4, 137.2, 136.4, 135.6, 134.8, 134.1, 134.0, 133.1, 133.0, 132.9, 132.8, 127.4, 126.9, 124.3, 121.4, 28.7 ppm; HRMS (+) *m*/*z* [M + H]^+^ 480.0688 (calcd for C_23_H_18_N_3_O_5_S_2_, 480.0679).

##### 9,10-Dioxo-*N*-((4-(N-(Pyrimidin-4-yl)Sulfamoyl)Phenyl)Carbamothioyl)-9,10-Dihydroanthracene-2-Carboxamide (**5j**)

Faint yellow (yield 70%), m.p. 237.0–237.2 °C; ^1^H NMR (300 MHz, DMSO-*d*_6_) *δ* 12.57 (s, 1H), 12.22 (s, 1H), 8.70 (d, *J* = 1.8 Hz, 1H), 8.53 (d, *J* = 4.9 Hz, 2H), 8.40 (dd, *J* = 8.1, 1.9 Hz, 1H), 8.32 (d, *J* = 8.1 Hz, 1H), 8.28–8.20 (m, 2H), 8.04 (d, *J* = 8.9 Hz, 2H), 7.98 (d, *J* = 2.3 Hz, 2H), 7.95 (q, *J* = 4.1, 3.4 Hz, 2H), 7.07 (t, *J* = 4.9 Hz, 1H), 2.08 (s, 1H) ppm; ^13^C NMR (75 MHz, DMSO-*d*_6_) *δ* 182.0, 181.8, 179.0, 166.7, 158.4, 158.3, 156.8, 141.8, 137.2, 135.6, 134.8, 134.1, 133.1, 133.1, 133.0, 132.8, 132.7, 128.3, 127.4, 127.0, 126.9, 124.5, 123.8, 110.1 ppm; HRMS (+) *m*/*z* [M + H]^+^ 544.0749 (calcd for C_26_H_18_N_5_O_5_S_2_, 544.0740).

##### ((4-(*N*-Acetylsulfamoyl)Phenyl)Carbamothioyl)-9,10-Dioxo-9,10-Dihydroanthracene-2-Carboxamide (**5k**)

Faint yellow solid (yield 65%), m.p. 192.0–192.4 °C; ^1^H NMR (300 MHz, DMSO-*d*_6_) *δ* 12.60 (s, 1H), 12.20 (d, *J* = 31.8 Hz, 2H), 8.71 (dd, *J* = 1.8, 0.6 Hz, 1H), 8.41 (dd, *J* = 8.1, 1.9 Hz, 1H), 8.32 (dd, *J* = 8.2, 0.5 Hz, 1H), 8.29–8.20 (m, 2H), 8.06–7.99 (m, 2H), 7.98 (d, *J* = 2.4 Hz, 2H), 7.97–7.93 (m, 2H), 1.95 (s, 3H) ppm; ^13^C NMR (75 MHz, DMSO-*d*_6_) *δ* 182.0, 181.8, 179.1, 168.8, 166.7, 142.5, 137.1, 136.2, 135.6, 134.8, 134.1, 133.1, 133.0, 132.8, 132.7, 128.4, 127.4, 127.4, 126.9, 124.0, 123.3, 23.3 ppm; HRMS (+) *m*/*z* [M + H]^+^ 508.0637 (calcd for C_24_H_18_N_3_O_6_S_2_, 508.0629).

##### *N*-((4-(N-(6-Methylpyrimidin-4-yl)Sulfamoyl)Phenyl)Carbamothioyl)-9,10-Dioxo-9,10-Dihydroanthracene-2-Carboxamide (**5l**)

Yellow solid (yield 74%), m.p. 240.0–240.5 °C; ^1^H NMR (300 MHz, DMSO-*d*_6_) *δ* 8.70 (d, *J* = 1.8 Hz, 1H), 8.40 (dd, *J* = 8.1, 1.8 Hz, 1H), 8.36–8.30 (m, 2H), 8.29–8.21 (m, 2H), 8.03 (d, *J* = 8.8 Hz, 2H), 7.98 (d, *J* = 3.4 Hz, 2H), 7.96 (d, *J* = 3.3 Hz, 2H), 6.92 (d, *J* =5.2 Hz, 1H), 2.33 (s, 3H)ppm; ^13^C NMR (75 MHz, DMSO-*d*_6_) *δ* 182.0, 181.8, 179.0, 166.7, 163.1, 157.5, 156.4, 141.6, 137.1, 135.6, 134.8, 134.1, 133.1, 133.0, 132.8, 132.7, 128.5, 127.4, 127.3, 126.9, 126.8, 124.7, 123.7, 114.8, 23.2 ppm; HRMS (+) *m*/*z* [M + H]^+^ 558.0906 (calcd for C_27_H_20_N_5_O_5_S_2_, 558.0892).

##### 9,10-Dioxo-*N*-((4-(N-(Thiazol-2-yl)Sulfamoyl)Phenyl)Carbamothioyl)-9,10-Dihydroanthracene-2-Carboxamide (**5m**)

Brown-yellow solid (yield 70%), m.p. 167.5–167.8 °C; ^1^H NMR (300 MHz, DMSO-*d*_6_) *δ* 12.54 (s, 1H), 8.72–8.69 (m, 1H), 8.40 (dd, *J* = 8.1, 1.9 Hz, 1H), 8.34–8.29 (m, 1H), 8.28–8.21 (m, 2H), 8.00–7.95 (m, 2H), 7.95–7.89 (m, 2H), 7.89–7.85 (m, 2H), 7.85–7.77 (m, 1H), 7.28 (d, *J* = 4.6 Hz, 1H), 6.86 (d, *J* = 4.6 Hz, 1H) ppm; ^13^C NMR (75 MHz, DMSO-*d*_6_) *δ* 182.0, 181.8, 179.1, 170.0, 166.7, 141.1, 137.2, 135.6, 134.8, 134.1, 133.1, 133.0, 132.8, 128.5, 128.4, 127.4, 126.9, 126.5, 124.5, 124.5, 124.3, 117.4, 108.3 ppm; HRMS (+) *m*/*z* [M + H]^+^ 549.0361 (calcd for C_25_H_17_N_4_O_5_S_3_, 549.0349).

##### ((4-(*N*-Carbamimidoylsulfamoyl)Phenyl)Carbamothioyl)-9,10-Dioxo-9,10-Dihydroanthracene-2-Carboxamide (**5n**)

Light yellow crystal (yield 65%), m.p. 206.0–206.5 °C; ^1^H NMR (300 MHz, DMSO-*d*_6_) *δ* 12.53 (s, 1H), 12.20 (s, 1H), 8.70 (d, *J* = 1.8 Hz, 1H), 8.41 (dd, *J* = 8.1, 1.9 Hz, 1H), 8.32 (d, *J* = 8.1 Hz, 1H), 8.29–8.21 (m, 2H), 7.97 (dd, *J* = 5.8, 3.3 Hz, 2H), 7.86 (d, *J* = 8.7 Hz, 2H), 7.80 (d, *J* = 8.6 Hz, 2H), 6.75 (s, 2H), 7.78 (s, 2H) ppm; ^13^C NMR (75 MHz, DMSO-*d*_6_) *δ* 182.0, 181.9, 179.1, 166.8, 158.1, 140.3, 137.2, 135.6, 134.8, 134.1, 133.9, 133.1, 133.0, 132.8, 132.8, 127.4, 127.2, 126.9, 126.2, 124.1, 123.5 ppm; HRMS (+) *m*/*z* [M + H]^+^ 508.0749 (calcd for C_23_H_18_N_5_O_5_S_2_, 508.0735).

#### 3.1.5. General Procedure for Target Compounds **6a**–**e** and **6h**–**k**

Raw materials **4a**–**e** and **4h**–**k** (1 mmol) were weighed and dissolved in 20 mL of dichloromethane/tetrahydrofuran; N,N-diisopropylethylamine (DIPEA, 1 mmol) was added and stirred in an ice bath for 15 min. Then, 1.5 mmol of anthraquinone-2-formyl chloride (**2**) was added, the mixture was stirred under ice-cold water for **6 h**, and the reaction was monitored by TLC. After the reaction was terminated, the precipitate was collected and recrystallized by dimethylformamide to obtain compounds **6a**–**e** and **6h**–**k**.

##### 4-(9,10-Dioxo-9,10-Dihydroanthracene-2-Carboxamido)Benzoic Acid (**6a**)

Yellow solid (yield 59%), m.p. 338.0–338.5 °C; ^1^H NMR (300 MHz, DMSO-*d*_6_) *δ* 10.97 (s, 1H), 8.76 (d, *J* = 1.8 Hz, 1H), 8.48–8.39 (m, 1H), 8.39–8.30 (m, 1H), 8.30–8.25 (m, 1H), 8.23 (dd, *J* = 5.4, 3.2 Hz, 1H), 7.99 (d, *J* = 4.2 Hz, 1H), 7.97 (s, 2H), 7.97 (s, 2H), 7.94 (d, *J* = 3.9 Hz, 1H) ppm; ^13^C NMR (75 MHz, DMSO-*d*_6_) *δ* 182.2, 182.1, 167.0, 164.3, 142.8, 139.3, 134.9, 134.7, 134.5, 133.4, 133.1, 133.1, 132.8, 130.1, 127.1, 126.9, 126.9, 126.2, 126.1, 119.8 ppm; HRMS (+) *m*/*z* [M + H]^+^ 372.0872 (calcd for C_22_H_14_NO_5_, 372.0865).

##### (9,10-Dioxo-9,10-Dihydroanthracene-2-Carboxamido)Benzoic Acid (**6b**)

Orange solid (yield 63%), m.p. 274.0–274.6 °C; ^1^H NMR (300 MHz, DMSO-*d*_6_) *δ* 10.87 (s, 1H), 8.78 (d, *J* = 1.8 Hz, 1H), 8.49–8.43 (m, 2H), 8.34 (d, *J* = 8.1 Hz, 1H), 8.29–8.20 (m, 2H), 8.09 (m, 1H), 7.96 (dd, *J* = 5.8, 3.3 Hz, 2H), 7.72 (m, 1H), 7.51 (t, *J* = 7.9 Hz, 1H) ppm; ^13^C NMR (75 MHz, DMSO-*d*_6_) *δ* 182.1, 182.1, 166.5, 164.0, 139.5, 134.9, 134.7, 134.2, 133.7, 133.3, 133.1, 133.0, 132.6, 131.4, 128.7, 127.1, 126.9, 126.1, 124,1, 124.1, 121.5, 119.6 ppm; HRMS (+) *m*/*z* [M + H]^+^ 372.0872 (calcd for C_22_H_14_N_1_O_5_, 372.0862).

##### 9,10-Dioxo-*N*-(4-Sulfamoylphenyl)-9,10-Dihydroanthracene-2-Carboxamide (**6c**)

White solid (yield 77%), m.p. 341.0–341.3 °C; ^1^H NMR (300 MHz, DMSO-*d*_6_) *δ* 11.01 (s, 1H), 8.77 (d, *J* = 1.8Hz, 1H), 8.46 (dd, *J* = 8.1, 1.8 Hz, 1H), 8.36 (d, *J* = 8.1 Hz, 1H), 8.26 (m, 2H), 8.02 (s, 1H), 7.98 (d, *J* = 3.7 Hz, 2H), 7.96 (d, *J* = 3.3 Hz, 1H), 7.85 (d, *J* = 8.8 Hz, 2H), 7.33 (s, 2H) ppm; ^13^C NMR (75 MHz, DMSO-*d*_6_) *δ* 182.2, 182.1, 164.3, 141.7, 139.2, 135.0, 134.9, 134.9, 134.7, 133.4, 133.3, 133.1, 133.1, 127.2, 126.9, 126.6, 126.1, 123.2, 120.2 ppm; HRMS (+) *m*/*z* [M + H]^+^ 407.0702 (calcd for C_21_H_15_N_2_O_5_S, 407.0692).

##### 9,10-Dioxo-*N*-(3-Sulfamoylphenyl)-9,10-Dihydroanthracene-2-Carboxamide (**6d**)

Faint yellow solid (yield 70%), m.p. 283.0–283.2 °C; ^1^H NMR (300 MHz, DMSO-*d*_6_) *δ* 10.99 (s,1H), 8.79 (dd, *J* = 1.9, 0.5 Hz, 1H), 8.47 (dd, *J* = 8.1, 1.9 Hz, 1H), 8.42–8.32 (m, 2H), 8.29–8.22 (m, 2H), 8.03 (m, 1H), 8.00–7.93 (m, 2H), 7.64–7.57 (m, 2H), 7.43 (s, 2H) ppm; ^13^C NMR (75 MHz, DMSO-*d*_6_) *δ* 182.2, 182.1, 164.2, 144.6, 139.2, 139.1, 135.0, 134.9, 134.7, 134.6, 133.4, 133.1, 133.1, 129.5, 127.2, 126.9, 126.1, 124.7, 123.4, 121.2, 117.6 ppm; HRMS (+) *m*/*z* [M + H]^+^ 407.0702 (calcd for C_21_H_15_N_2_O_5_S_1_, 407.0691).

##### *N*-(4-Acetylphenyl)-9,10-Dioxo-9,10-Dihydroanthracene-2-Carboxamide (**6e**)

Light yellow crystal (yield 80%), m.p. 266.0–266.5; ^1^H NMR (300 MHz, DMSO-*d*_6_) *δ* 10.95 (s, 1H), 8.75 (s, 1H), 8.44 (d, *J* = 8.2 Hz, 1H), 8.33 (d, *J* = 8.1 Hz, 1H), 8.24 (s, 2H), 8.01 (s, 1H), 7.98 (s, 2H), 7.97–7.94 (m, 2H), 7.91 (s, 1H), 2.50 (s, 3H) ppm; ^13^C NMR (75 MHz, DMSO-*d*_6_) *δ* 197.1, 182,6, 182.6, 164.9, 143.6, 139.8, 135.5, 135.5, 135.2, 134.7, 133.9, 133.7, 133.6, 132.9, 129.8, 127.7, 127.4, 127.4, 126.6, 120.2, 27.0 ppm; HRMS (+) *m*/*z* [M + H]^+^ 370.1079 (calcd for C_23_H_16_NO_4_, 370.1069).

##### 9,10-Dioxo-*N*-(4-Sulfamoylphenethyl)-9,10-Dihydroanthracene-2-Carboxamide (**6h**)

White solid (yield 75%), m.p. 329.0–329.4 °C; ^1^H NMR (300 MHz, DMSO-*d*_6_) *δ* 9.09 (t, *J* = 5.5 Hz, 1H), 8.62 (t, *J* = 1.1 Hz, 1H), 8.28 (t, *J* = 1.3 Hz, 2H), 8.26–8.23 (m, 1H), 8.23–8.18 (m, 1H), 7.99–7.91 (m, 2H), 7.76 (d, *J* = 8.2 Hz, 2H), 7.50–7.42 (m, 2H), 7.30 (s, 1H), 3.58 (t, *J* = 6.5 Hz, 2H), 2.98 (t, *J* = 7.1 Hz, 2H) ppm; ^13^C NMR (75 MHz, DMSO-*d*_6_) *δ* 182.1, 182.1, 164.6, 143.7, 142.1, 139.2, 134.6, 134.5, 134.5, 133.1, 133.0, 132.7, 131.6, 129.2, 127.1, 126.8, 126.8, 126.7, 125.7, 125.4, 40.7, 34.6 ppm; HRMS (+) *m*/*z* [M + H]^+^ 435.1015 (calcd for C_23_H_19_N_2_O_5_S_1_, 435.1003).

##### *N*-(4-(N-Methylsulfamoyl)Phenyl)-9,10-Dioxo-9,10-Dihydroanthracene-2-Carboxamide (**6i**)

White solid (yield 60%). m.p. 273.0–273.8 °C; ^1^H NMR (300 MHz, DMSO-*d*_6_) *δ* 11.03 (s, 1H), 8.76 (dd, *J* = 1.9, 0.5 Hz, 1H), 8.45 (dd, *J* = 8.1, 1.9 Hz, 1H), 8.37–8.32 (m, 1H), 8.30–8.25 (m, 1H), 8.25–8.20 (m, 1H), 8.08–8.01 (m, 2H), 8.01–7.92 (m, 2H), 7.84–7.77 (m, 2H), 7.39 (q, *J* = 5.0 Hz, 1H), 2.43 (d, *J* = 5.0 Hz, 3H) ppm; ^13^C NMR (75 MHz, DMSO-*d*_6_) *δ* 182.1, 182.0, 164.4, 142.3, 139.2, 135.0, 134.9, 134.9, 134.7, 134.1, 133.9, 133.4, 133.1, 128.1, 127.7, 127.2, 126.9, 126.1, 120.3, 28.7 ppm; HRMS (+) *m*/*z* [M + H]^+^ 421.0858 (calcd for C_22_H_17_N_2_O_5_S, 421.0847).

##### 9,10-Dioxo-*N*-(4-(N-(Pyrimidin-4-yl)Sulfamoyl)Phenyl)-9,10-Dihydroanthracene-2-Carboxamide (**6j**)

Red crystal (yield 75%), m.p. 317.0–317.5 °C; ^1^H NMR (300 MHz, DMSO-*d*_6_) *δ* 11.77 (s, 1H), 11.03 (s, 1H), 8.76–8.72 (m, 1H), 8.52 (d, *J* = 4.9Hz, 2H), 8.43 (dd, *J* = 8.1, 1.9 Hz, 1H), 8.34 (d, *J* = 8.1 Hz, 1H), 8.24 (m, 2H), 8.01–7.99 (s, 4H), 7.96 (dd, *J* = 5.8, 3.3 Hz, 2H), 7.06 (t, *J* = 4.9 Hz, 1H) ppm; ^13^C NMR (75 MHz, DMSO-*d*_6_) *δ* 182.1, 182.0, 164.5, 158.4, 158.3, 157.0, 142.7, 139.2, 135.0, 134.7, 134.7, 133.4, 133.1, 133.1, 132.9, 132.8, 128.8, 127.2, 126.9, 126.1, 125.3, 119.9, 109.1 ppm; HRMS (+) *m*/*z* [M + H]^+^ 485.0920 (calcd for C_25_H_17_N_4_O_5_S, 485.0908).

##### *N*-(4-(N-Acetylsulfamoyl)Phenyl)-9,10-Dioxo-9,10-Dihydroanthracene-2-Carboxamide (**6k**)

White solid (yield 70%), m.p. 304.0–304.3 °C; ^1^H NMR (300 MHz, DMSO-*d*_6_) *δ* 11.08 (s, 1H), 8.78–8.74 (m, 1H), 8.45 (dd, *J* = 8.1, 1.9Hz, 1H), 8.35 (d, *J* = 8.1 Hz, 1H), 8.29–8.25 (m, 1H), 8.25–8.21 (m, 1H), 8.08–8.02 (m, 2H), 8.00–7.95 (m, 2H), 7.95–7.88 (m, 2H), 1.93 (s, 3H) ppm; ^13^C NMR (75 MHz, DMSO-*d*_6_) *δ* 182.1, 182.0, 168.8, 164.5, 143.3, 139.1, 135.0, 134.7, 134.0, 133.6, 133.4, 133.1, 133.1, 132.5, 128.7, 127.2, 126.9, 126.2, 124.8, 120.0, 23.3 ppm; HRMS (+) *m*/*z* [M + H]^+^ 449.0807 (calcd for C_23_H_17_N_2_O_6_S, 449.0797).

### 3.2. Biological Studies

#### 3.2.1. CA Inhibition Assay

CA presents a considerable degree of three-dimensional similarity and typical folds. The active site is located in a large cone-shaped cavity that reaches the center of the protein molecule. The active site contains a zinc ion bound to a hydroxyl ion (OH^−^). Additionally, in the CA, three histidine residues (His 94, His 96, and His 119) are present, and their side-chain residues form coordinate bonds with the zinc ion. Therefore, CA has the same catalytic activity and can be measured using the same method for enzyme activity [29].

On the one hand, CA can participate in the hydration reaction of CO_2_ [30], catalyzing the interconversion of CO_2_ and H_2_O with HCO_3_^−^ and H^+^. On the other hand, CA also has esterase activity [31], which can combine with CO_2_ to form a CA–carbonic acid complex and then transfer the H^+^ in the complex to the substrate, forming carboxylic acid and alcohol. In this process, CO_2_ plays the role of activating the substrate, making it easier to undergo hydrolysis. The catalytic mechanism and characteristics of CA are similar to those of esterase. Therefore, according to the method described by Verpoorte previously [32], **4-NPA** was catalyzed into 4-nitrophenol for the determination of CA activity.

The activity of CA is reflected by an increased absorbance value measured by an ultraviolet (UV)–visible spectrophotometer at a wavelength of 405 nm [33]. All compounds and the positive control AAZ were dissolved in DMSO at 25 °C and diluted to the indicated concentrations with 1× assay buffer (HEPES 7.2) [34]. Ten different compound and AAZ concentrations were utilized (30, 10, 3.3333, 1.1111, 0.3704, 0.1235, 0.0412, 0.0137, 0.0046, 0.0015, and 0 µM). The substrate (4-NPA) was dissolved in DMSO at a substrate concentration of 1 mM. Standard hCAs II and IX (acquired from Shanghai Jining Biological) were diluted to the indicated concentrations with assay buffer and added to the cells. Different concentrations of inhibitors were added to the cells, followed by incubation at room temperature for 15 min to form complexes of inhibitors and enzymes (E-I). Then, 100 μL of E-I complex and 900 μL of substrate were successively added to a 1 mL glass cuvette and mixed immediately. The absorbance values at 405 nm were measured at 3 min (A1) and at 6 min (A2), and the change in absorbance value was calculated using the following formula: ΔA = A2 − A1. At the same time, the non-enzymatic hydrolysis rate was determined by a blank control test. Standard hCAs II and IX were diluted to final concentrations of 1.5 ng/μL and 5 ng/μL, respectively [26,35]. The FORECAST (x, known-y’s, known-x’s) command of Excel was used to return a predicted value through the linear regression equation to calculate the IC_50_ values. The experiments were repeated three times for each inhibitor concentration.

#### 3.2.2. Cell Lines

The MDA-MB-231, MCF-7, and HepG2 cell lines were provided by the School of Pharmacy, Jilin University, China. The cells were incubated with 5% CO_2_ at 37 °C and then cultured in RPMI-1640 medium with 10% (*v*/*v*) fetal bovine serum (FBS). The cell cultures were passaged once every 2–3 days.

#### 3.2.3. MTT Assay

The 3-(4,5-dimethylthiazol-2-yl)-2,5-diphenyltetrazolium bromide (MTT) assay was used to measure the cell viability. Cells were planted into 96-well plates (4 × 10^3^ cells per well) and cultured in media containing different concentrations of compounds. After incubation for 24 h, 100 μL of medium containing 5 mg/mL MTT was added to each well, followed by incubation for 4 h at 37 °C in the dark. The supernatant was then discarded, and 150 μL of DMSO was added to each well, followed by gentle shaking. The absorbance value at 570 nm was measured using an EL808 microplate reader (Bio-TEK Instrument, Winooski, VT, USA). The test was repeated in triplicate, and the IC_50_ values were calculated by Excel software (Microsoft Office 2021 LTSC). The hypoxic atmosphere consisted of 0.5% O_2_, 5% CO_2_, and 94.5% N_2_, while the normoxic atmosphere consisted of 20% O_2_, 5% CO_2_, and 75% N_2._

#### 3.2.4. Cell Apoptosis Assay

Cells were cultured in 6-well plates and treated with the compounds for 24 h. Approximately 1 × 10^5^ cells were collected and suspended in binding buffer. Apoptosis was analyzed using an AV/PI apoptosis detection kit (Beyotime Institute of Biotechnology, Shanghai, China) with a flow cytometer (BD Bioscience, Franklin Lakes, NJ, USA).

#### 3.2.5. Mitochondrial Membrane Potential Assay

A mitochondrial membrane potential assay kit (Beyotime Co., Nantong, China) was used to measure the mitochondrial membrane potential with a JC-1 fluorescent probe. Approximately 5 × 10^5^ cells were incubated with different concentrations of the compounds for 24 h in 6-well plates. The cells were then fixed with 4% paraformaldehyde and stained with 5 mg/L JC-1 at 37 °C for 1 h. Fluorescent images were acquired using a fluorescence microscope (Olympus IX81, Olympus, Tokyo, Japan).

### 3.3. Molecular Modeling Studies

The crystal structures of hCA II (PDB ID: 5GMN) and hCA IX (PDB ID: 5FL4) were downloaded from the Protein Data Bank. The SMILES strings of each compound were converted into a 3D SDF format using Open Babel software (version 3.1.1) [36]. The PDB files were converted to PDBQT using AutoDock Tools (version 1.5.7p1 Nov_2_22) to make them compatible with AutoDock Vina (version 1.2.5). Ligands were prepared for the PDBQT files with the Meeko Python package (version 0.5.0). The x, y, and z axes of the grid box were divided into 60 grid points, and the grid point spacing was 0.375Å. The grid center coordinates of hCA II (PDB ID: 5GMN) were set to X = 9.834, Y = 2.147, Z = 12.822. The grid center coordinates of hCA IX (PDB ID: 5FL4) were set to X = 14.370, Y = −27.413, Z = 59.845. A specialized force field, the AutoDock4 Zn force field [37], was applied to the zinc ions of these zinc metalloenzymes by adding tetrahedral zinc pseudo-atoms. The molecular docking was performed using AutoDock Vina [38], and AD4 scoring was used to elucidate the binding mechanisms of CA II (PDB ID: 5GMN) and CA IX (PDB ID: 5FL4) with the compounds. The docking results were visualized using PyMOL software (version 2.5.0), and the interactions were identified using PLIP software (version 2.3.0) [39].

### 3.4. ADME Studies

Absorption, distribution, metabolism, and excretion (ADME) properties were evaluated using the SwissADME [40,41] online software (http://www.swissadme.ch, accessed on 20 November 2023).

## 4. Conclusions

In the present work, two novel series of anthraquinone-based sulfonamide derivatives and analogues (compounds **5a**–**n**, **6a**–**e**, and **6h**–**k**) were designed, synthesized, and screened for their inhibitory activity against two CA isoforms, namely, off-target hCA II, and the tumor-related hCA IX. Most of the newly reported compounds efficiently inhibited hCA II and hCA IX, with IC_50_ values in the 21.19–295.94 nM and 30.06–274.81 nM ranges, respectively. SAR analysis revealed that the primary anthraquinone-based sulfonamides (compounds **5c**–**d**, **5h**, **6c**–**d**, and **6h**) exerted better activity than their carboxylic acid (compounds **5a**, **5b**, **6a**, and **6b**), ketone (compounds **5e**, and **6e**), and ester (compounds **5f**, and **5g**) analogues, and shifting the primary sulfonamido functionality from the *para*- to the *meta*-position led to a decrease in the hCA-inhibitory activities. In addition, compounds **5c**, **5h**, **6c**, and **6h** efficiently inhibited the tumor-related hCA IX isoform (IC_50_ = 40.58, 46.19, 34.88, and 30.06 nM, respectively). Furthermore, compounds **5c**, **5h**, **6c**, and **6h** were assessed for their potential anticancer activities against the MDA-MB-231, MCF-7, and HepG2 cancer cell lines, and compound **5c** showed moderate anti-proliferative activity. Apoptosis and mitochondrial membrane potential assays were used to demonstrate that compound **5c** promoted cancer cell apoptosis. Moreover, a molecular docking study showed the relationship between the structural features and inhibitory profile against the off-target hCA II and the tumor-related hCA IX isoforms.

Due to their unique chemical structure and biological activity, anthraquinone compounds occupy a significant position in drug development. In this study, we synthesized and screened anthraquinone compounds that exhibited good inhibitory activity against carbonic anhydrase, providing additional options for the design and development of CA inhibitors. Through continuous exploration and research, we aim to discover more effective natural products that can be used to develop more potent and safer carbonic anhydrase inhibitors.

## Data Availability

Data is contained within the article and Appendix A.

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
