# Peer review of "Novel Anthraquinone-Based Benzenesulfonamide Derivatives and Their Analogues as Potent Human Carbonic Anhydrase Inhibitors with Antitumor Activity: Synthesis, Biological Evaluation, and In Silico Analysis"

_ijms, 2024, doi:10.3390/ijms25063348_

Round 1
Reviewer 1 Report
Comments and Suggestions for Authors
IJMS-2867233: Reviewer comments
The submitted manuscript reports two series of novel anthraquinone-based benzenesulfonamide derivatives and their analogues as potential carbonic anhydrase IX (CAIX) inhibitors potentially useful for the treatment of cancer. The inhibitory activities of the synthesized compounds were measured against the tumor-associated hCAIX and also off-targeted hCA II isoform. The anticancer activities of some selected inhibitors were evaluated on suitable cancer cell lines. These studies let emerged compound 5c as endowed with good anti-tumor activity under both normoxic and hypoxic conditions. Additional apoptosis and mitochondrial membrane potential assays, molecular docking studies, and ADME predictions were also reported.
The results reported in the submitted manuscript seem to have some interest and add some knowledge to the research in this field. Thus the manuscript warrants consideration for publication as an article in the IJMS but only after revision. Some major and minor issues are listed below:
1) Throughout the maniscript, the numbers and letters indicating chemical entities (target compounds, intermediates or others cited in the work) should be reported in BOLD character.
2) Page 4.
- Section “Results”. Lacking the research manuscript section “Discussion”, this section should be termed “Results and Discussion”.
- Figure 2 has to be enlarged as much as possible. Molecular structures of the reported compounds and also their corresponding names/abbreviations/numbers are too small.
3) Page 5.
- Section “Chemistry”. Discussion on NMR spectral data should be deleted due to the simplicity and unicity of the synthetic procedures and structures of the target compounds. If you think it is important, replace lines 1-19 with "NMR spectal data are in accordance with the structures of the reported compounds".
- Section “Biological evaluation”. The inhibitory activity was evaluated by monitoring the hydrolisis of 4-NPA. This method, based on the measurement of the CA esterase activity, was discovered in the early 1960s ans has now been superseded by the stopped-flow measurements of CO2 idratation inhibition. However, despite the relatively slow kinetics compared to the physiologically relevant hydratase activity, esterase activity can be considered indicative of hydratase activity. In fact, several structural, functional and mutational studies have shown that the two distinct enzymatic activities of CA share similar mechanism and the same catalytic pocket. These information should be reported in the manuscript with suitable bibliographical references. The manuscript could also be read by researchers not expert in this field, and the correlation between esterase and hydratase acivity of CA should be reported.
Moreover, to improve the quality of the manuscript, I suggest also evaluating the inhibitory activity towards the tumor-associated hCAXII isoform which is highly expressed in many human cancers and has important roles in carcinogenesis and tumor progression. Compounds with good inhibitory activity on both CA IX and XII isoforms could have a synergisting action in combating cancer.
- Table 1. It should be reported in full on the same page and not split between two subsequent pages. In column 2 (m- / p-) the term “P-“ (para-) has to be reported in lowercase font as the term m- (meta-).
Comments on the Quality of English LanguageMinor editing of English language is required throughout the manuscript.
Reviewer 2 Report
Comments and Suggestions for Authors
The article submitted by D. Liang describe synthesis, biological evaluation, and in silico analysis of anthraquinone–based benzenesulfonamide derivatives. The article is well-written and looks interesting for the readers. However, there are a lot of major issues especially in experimental part and some minor issues in text. The revision is necessary.
#1 Experimental part. Carbon spectra should be round to one digit after dot.
#2 Experimental part. MS. Authors used the ESI-MS but result the reported results have only one digit after dot. To confirm the purity of obtained compounds the elemental analysis of HRMS with four digits after dot should be used.
#3 All compound names should be started with capital letter.
#4 In the experimental part there are a lot of compounds with less found protons than they actually have. It really might be with compounds have labile NH or COOH protons. Authors should carry out the spectra in different solvent (if possible) or with addition of СF3COOD acid in DMSO-d6. Authors should also add notes to problematic compounds with explanation which protons are missed.
· Experimental part. 5a has 14H in 1H NMR only 13H found.
· Experimental part. 5b has 14H in 1H NMR only 13H found.
· Experimental part. 5c has 15H in 1H NMR only 13H found.
· Experimental part. 5d has 15H in 1H NMR only 12H found.
· Experimental part. 5e has 16H in 1H NMR only 14H found.
· Experimental part. 5f has 16H in 1H NMR only 12H found.
· Experimental part. 5g has 16H in 1H NMR only 15H found.
· Experimental part. 5h has 19H in 1H NMR only 18H found.
· Experimental part. 5i has 17H in 1H NMR only 15H found.
· Experimental part. 5l has 19H in 1H NMR only 16H found.
· Experimental part. 5m has 16H in 1H NMR only 15H found.
· Experimental part. 5n has 17H in 1H NMR only 15H found.
· Experimental part. 6a has 13H in 1H NMR only 12H found.
· Experimental part. 6b has 13H in 1H NMR only 12H found.
· Experimental part. 6h has 18H in 1H NMR only 17H found.
· Experimental part. 6j has 16H in 1H NMR only 13H found.
· Experimental part. 6k has 16H in 1H NMR only 15H found.
#5 All compounds number should be in bold style.
#6 Some Figures in text have a blue color and underlined some just underlined. Please explain the difference or format all equally.
#7 Abstract. Abstract is too long. Abstract should summarize the result of work, i.e. first sentence is abundant. It should be avoided to use the compound number in abstract (sentence started with “Compound 5c”)
#8 Figure 2. The anthrane-9,10-dione has a blue spot, please explain or remove this. The commas are looked unreadable. Find the picture attached.

#9 The sentence started with “In the present study, an-thraquinone derivatives were….” Should be as new paragraph.
#10 Same sentence “including primary sulfamoyl (compounds 5c-d, 5h ,6c-d, and 6h), secondary sulfamoyl (compounds 5i-n and 6i-k), free carboxylic acid (compounds 5a-b and 6a-b), acetyl (compounds 5e and 6e), and ethyl ester (compounds 5f-g)” Authors obviously wanted to describe the aim of research but they wrote about obtained result. The sentence should be rewritten.
#11 The several of next sentences are also described the result and should not be in the introduction section. Find the attach below.

#12 Results. Chemistry. Paragraph 2 sentence 2. “was refluxed in the presence of acetone solution and potas-sium thiocyanate to produce the key….” should be “was refluxed in acetone with the presence of potassium thiocyanate to produce the key…..”
#13 The Scheme 1 and 2 are very similar. It should be more readable if Authors merge them into one Scheme.
#14 The discussion after Table 1, the (i) (ii) (iii) etc is quite confused and overlap with Schemes, please remove the mark or replace it by number or by other marks.
#15 ESI. Contest. Specify name the Figure S1 & Table S2 and S3.
#16 ESI. Contest. MS data. There are no MS data in ESI, only NMR spectra and docking result. Please, check.

Reviewer 3 Report
Comments and Suggestions for Authors
The manuscript presents the synthesis of two novel series of anthraquinone-based benzenesulfonamides and evaluates their inhibitory activities against CA II and IX, alongside their anticancer activity under normoxic and hypoxic conditions. I have to mention positively, that they chose suitable CA expressing cancer cell lines and studied them under normoxic as well as hypoxic conditions. They support their findings well by docking and ADME studies. The writing is clear and coherent. Overall, the manuscript presents valuable findings and contributes to the understanding of anthraquinone-based compounds as potential therapeutics, and with some minor revisions, it has the potential to be a interesting contribution to the field.
Recommendations for improvement:
- Increase the resolution of Figures 1, 2, 4, and 5.
- Explain why only the two isoforms were used.
- Correct the typo in Figure 4: "Aneexin"
- Outline potential future directions for research based on the findings, such as further structural optimization of lead compounds, in vivo efficacy studies and so on
- Ensure full assignment of NMR data.
Round 2
Reviewer 2 Report
Comments and Suggestions for Authors
I'm pleased how Authors have rewritten and worked up with manuscript. However, the HRMS or elemental analysis is still necessary for accept the publication.
Authors wrote in Answert to Referee: "Thank you for the above suggestions. The high-resolution mass spectrometry (HRMS) measurements for the 23 compounds are currently being conducted and will be completed as soon as possible. Upon completion, the relevant data and spectra will be included in the manuscript and supplementary materials." Please, upload an updated manuscript as you finished all of your additional experiments, you may ask Editor for providing additional times.
Author Response
We have added HRMS data as requested by the reviewers and editors. The HRMS data has been incorporated into the main text and supporting information.
Round 3
Reviewer 2 Report
Comments and Suggestions for Authors
I'm pleased how Authors have taken into account the reviewer's suggestion. I'd like to recommend to publish the manuscript in IJMS.
Author Response
Thank you for your advice and guidance.